

# Expression, localization, and function of P4HB in the spermatogenesis of Chinese mitten crab (*Eriocheir sinensis*)

Yulian Tang*, Anni Ni*, Shu Li, Lishuang Sun and Genliang Li

Youjiang Medical University for Nationalities, Baise, Guangxi, China
* These authors contributed equally to this work.

## ABSTRACT

**Background:** The sperm of Chinese mitten crab (*Eriocheir sinensis*) have special noncondensed nuclei. The formation and stability of the special nuclei are closely related to the correct folding of proteins during spermatogenesis. P4HB plays a key role in protein folding, but its expression and role in the spermatogenesis of *E. sinensis* are unclear.

**Objective:** To investigate the expression and distribution characteristics of P4HB in the spermatogenesis of *E. sinensis* as well as its possible role.

**Methods:** The testis tissues of adult and juvenile *E. sinensis* were used as materials. We utilized a variety of techniques, including homology modeling, phylogenetic analysis, RT-qPCR, western blotting, and immunofluorescence staining to predict the protein structure and sequence homology of P4HB, analyze its expression in the testis tissues, and localize and semi-quantitatively assess its expression in different male germ cells.

**Results:** The sequence of P4HB protein in *E. sinensis* shared a high similarity of 58.09% with the human protein disulfide isomerase, and the phylogenetic tree analysis indicated that the protein sequence was highly conserved among crustaceans, arthropods, and other animals species. P4HB was found to be expressed in both juvenile and adult *E. sinensis* testis tissues, with different localization patterns observed all over the developmental stages of male germ cells. It was higher expressed in the spermatogonia, spermatocytes, and stage I spermatids, followed by the mature sperm than in the stage II and III spermatids. The subcellular localization analysis revealed that P4HB was predominantly expressed in the cytoplasm, cell membrane, and extracellular matrix in the spermatogonia, spermatocytes, stage I and stage II spermatids, with some present in specific regions of the nuclei in the spermatogonia. In contrast, P4HB was mainly localized in the nuclei of stage III spermatids and sperm, with little expression observed in the cytoplasm.

**Conclusion:** P4HB was expressed in the testis tissues of both adult and juvenile *E. sinensis*, but the expression and localization were different in male germ cells at various developmental stages. The observed differences in the expression and localization of P4HB may be an essential factor in maintaining the cell morphology and structure of diverse male germ cells in *E. sinensis*. Additionally, P4HB expressed in the nuclei of spermatogonia, late spermatids, and sperm may play an indispensable role in maintaining the stability of the noncondensed spermatozoal nuclei in *E. sinensis*.

Corresponding author
Genliang Li, ligenliang@163.com

# INTRODUCTION

Spermatogenesis is a complex physiological process that includes spermatogonia proliferation and differentiation into spermatocytes, spermatocyte meiosis resulting in the production of spermatids, and the metamorphosis of spermatids into sperm (*Nishimura & L'Hernault, 2017*). Chinese mitten crab (*Eriocheir sinensis*, *E. sinensis*) spermatogenesis differs significantly from that of other mammals. In *E. sinensis*, the spherical-shaped spermatid nucleus is gradually depressed and squeezed into the one side of the cell and forms a thin, cup-shaped spermatozoal nucleus that surrounds half of the acrosome as the proacrosomal vesicle gradually enlarges during the later stage of spermatogenesis (*Xu et al., 2018*; *Li et al., 2017*). The cup-shaped spermatozoal nucleus is susceptible to instability, compared to the stable, round or oval-shaped nuclei in cells (*Chen et al., 2020*; *Wu et al., 2016*). The former appears in the sperm of many decapod crustaceans, such as *E. sinensis* and is named noncondensed nucleus with genetic material in the form of fibrous chromatin filament (*Wei et al., 2019*; *Wu et al., 2015*) while the latter presents in the sperm of most animals and is named condensed nucleus. At present, the formation and stability mechanisms of these unique spermatozoal nuclei in *E. sinensis* remain unclear.

Cytoskeleton-related proteins are closely associated with the formation and stability of the noncondensed spermatozoal nuclei in *E. sinensis* (*Li et al., 2022*; *Wang et al., 2023*). *E. sinensis* is able to maintain the integrity of its unique spermatozoal nuclei and passes on genetic material to next generations, indicating the pivotal role of cytoskeleton-related proteins which can stabilize the cellular or subcellular morphology and structure. Prolyl 4-hydroxylase beta polypeptide (P4HB) is a cytoskeleton-related protein (*Sobierajska et al., 2014*). This abundant multifunctional enzyme is known to be in the endoplasmic reticulum, acting as a subunit of the microsomal triglyceride transporter protein. P4HB catalyzes the formation of 4-hydroxyproline in collagen and other collagen-like proteins, and plays a key role in facilitating the folding of many newly synthesized proteins as a chaperone-like multifunctional peptide (*Kari & Johanna, 1998*; *Wu et al., 2021*). Protein disulfide isomerase (PDI) is a multifunctional protein located in the lumen of the rough endoplasmic reticulum. It catalyzes the redox and isomerization of disulfide bonds, promoting the correct folding of proteins (*Li et al., 2018*). P4HB gene in humans primarily encodes PDI, which is a chaperone protein responsible for the folding of spermatocyte-specific protein present in the testis (*Li et al., 2018*; *Wu et al., 2021*). Several studies have reported PDI's presence in the testes of other mammals (*e.g.*, rat, mouse, rabbit, sheep), implicating its involvement in crucial processes such as spermatogenesis and sperm-egg fusion (*Akama et al., 2010*; *Lv et al., 2011*; *Zhao et al., 2013*).

However, the role of P4HB in regulating spermatogenesis in decapod crustaceans has not been reported yet. Therefore, we used testis tissues from adult and juvenile *E. sinensis*

crabs to analyze the expression and distribution characteristics of P4HB through RT-qPCR, western blotting, and immunofluorescence staining techniques. Additionally, we also explored the possible functional roles of P4HB in the *E. sinensis* spermatogenesis. Our study could provide a theoretical foundation for comprehending the molecular mechanisms of spermatogenesis and reproductive regulation in *E. sinensis*. These results can also serve as a valuable reference for species conservation and aquaculture management of *E. sinensis*.

## MATERIALS AND METHODS

### Materials

#### Experimental animals

Chinese mitten crabs were purchased from Ice King Aquatic Products Sales Co., Ltd. (Suqian, China). Thirty adult crabs weighing between 100–150 g and thirty juvenile crabs weighing between 20–30 g, with complete limbs and uniform size and vitality were selected, respectively. After acquisition, they were temporarily housed for 1 day under fully aerated tap water (water temperature 24 °C, pH 6.8, and dissolved oxygen levels above 5.0 mg/L) and subjected to testis tissue isolation and extraction. All animal procedures were performed at the Experimental Animal Center of Youjiang Medical University for Nationalities, and the study was conducted in accordance with national animal welfare regulations.

#### Main reagents and instruments

The Total RNA Extraction Kit was purchased from Solabo Technology Co., Ltd. (Beijing, China). The First-Strand cDNA Synthesis Kit was purchased from Thermo Fisher Technology Co., Ltd. (Shanghai, China). Blaze Taq SYBR® Green qPCR mix was purchased from iGine Biotechnology Co., Ltd. (Guangzhou, China). The Roche LightCycler 96 PCR instrument was purchased from F. Hoffmann-La Roche Co., Ltd. (Basel, Switzerland). The ultra-sensitive multifunctional imager was purchased from Cytiva Co., Ltd. (Marlborough, America), and the laser confocal microscope was purchased from Olympus Co., Ltd. (Tokyo, Japan).

P4HB and *β-Actin* primers were purchased from Shenggong Bioengineering Co., Ltd. (Shanghai, China). Anti-P4HB antibody [EPR949] (ab137110), Anti-beta Actin antibody [mAbcam 8226] (ab8226), Anti-DDX4/MVH antibody [mAbcam27591] (ab27591), Goat Anti-Mouse IgG H&L (HRP) (ab6789), Goat Anti-Mouse IgG H&L (Alexa Fluor® 594) (ab150116), Goat Anti-Rabbit IgG H&L (HRP) (ab6721), and Goat Anti-Rabbit IgG H&L (Alexa Fluor® 488) (ab150077) were all purchased from the same company, Abcam Trading Co., Ltd. (Shanghai, China).

### Methods

#### RNA extraction and sequencing

Approximately 25 mg of testis tissue was collected was taken from each adult and juvenile crab, and every 4–5 crab samples from the same group were pooled to obtain a sample of approximately 100 mg. Each group, including adult and juvenile crabs, had three

biologically replicates ($n = 3$). The samples were immediately flash-frozen in liquid nitrogen and ground to a fine powder. Total RNA extraction was performed on each sample using the Total RNA Extraction Kit, and the resulting RNA solution was considered qualified when the 260/280 ratio was between 1.8–2.0 and the 260/230 ratio was greater than 2.0. The purified RNA samples were sent to Shenzhen BGI Gene Co., Ltd. for RNA-seq. Poly(A) mRNA was isolated using the NEBNext Poly(A) mRNA Magnetic Isolation Module (NEB), mRNA fragmentation and priming were performed using the NEBNext First Strand Synthesis Reaction Buffer and NEBNext Random Primers. The libraries were then multiplexed with different indices and loaded onto an Illumina HiSeq instrument (Illumina, San Diego, CA, USA). Sequencing was carried out using a 2 × 150 bp paired-end (PE) configuration, followed by image analysis and base calling by the HiSeq instrument. Lastly, the *P4HB* mRNA sequence was obtained by comparing the genome sequence of *E. sinensis*.

### P4HB protein tertiary structure prediction and phylogenetic tree construction

The *P4HB* mRNA sequence was analyzed using the Expasy Online website (https://web. expasy.org/translate/) to obtain the corresponding amino acid sequence. Subsequently, the tertiary structure model of this protein was constructed using the SWISS-MODEL homology modeling website (https://swissmodel.expasy.org/interactive). Additionally, sequence comparison was performed using the NCBI-BLAST online tool (https://blast. ncbi.nlm.nih.gov/Blast.cgi), and the MAGA 7.0 software was used for sequence analysis. Finally, the evolutionary history of P4HB was investigated using neighbor-joining (NJ) analysis, and a phylogenetic tree was generated.

### P4HB mRNA expression analysis using RT-qPCR

Total RNA from testis tissues of both adult and juvenile crab samples, which were pooled from every 4–5 crabs in the same group, was reverse transcribed into cDNA. Real-time quantitative PCR was then performed using a fluorescent probe. The reaction conditions were set as follows: pre-denaturation at 95 °C for 30 s, followed by 40 cycles of denaturation at 95 °C for 5 s, and annealing/extension at 60 °C for 30 s. The dissolution curve reaction was conducted as per the recommendations of the Roche LightCycler96 instrument. The relative expression of *P4HB* mRNA between adult and juvenile crab samples was calculated using the $2^{-\Delta\Delta Ct}$ method.

### P4HB protein expression analysis using western blotting

Testis tissues were obtained from both adult and juvenile crab samples (200 mg each) and rapidly ground into a fine powder using liquid nitrogen. The tissue lysate was then added to each sample (1 ml of lysate per 50–100 mg of powder), mixed thoroughly, and placed on ice for 5 min before being ground using a tissue grinder. The total protein solution was then obtained by centrifugation at 12,000 r/min for 5 min at 4 °C. The concentration of total protein was determined using the BCA method, and 5× loading buffer was added to adjust the protein solution to the same concentration. The mixture was then heated and boiled for 5 min to obtain a denatured protein. SDS-PAGE and Western blotting were conducted according to the method proposed by *Pang et al. (2021)*. The primary antibody

was used at a 1:100 dilution, while the second antibody was used at a 1:5,000 dilution. The membranes were then imaged using an ultra-sensitive multifunctional imager and quantitatively analyzed using Image J software.

### P4HB protein expression analysis using immunofluorescence localization

Fresh testis tissues from both adult and juvenile crabs were immediately fixed using formaldehyde and subsequently made into paraffin sections. The sections were then dewaxed, rehydrated, and subjected to antigen retrieval. Primary antibodies were added to the slices at a 1:100 dilution and incubated overnight at 4 °C. Subsequently, fluorescent secondary antibodies were added at a 1:5,000 dilution and incubated in the dark for 1 h. The slices were then stained with a fluorescent cell nuclear dye and observed and photographed using a confocal microscope. The average fluorescence intensity in different cells was analysed using ImageJ software. The formula used for calculating the average fluorescence intensity was: average fluorescence intensity = integrated density in the region/area of the region.

## Statistical analysis

A statistical analysis of the data was performed using SPSS 13.0 statistical software. The results were expressed as mean ± standard deviation. After testing for normality and the chi-square test, a two-samples independent t-test was used to compare the two types of samples that conformed to a normal distribution. For multiple samples that conformed to a normal distribution, a completely randomized ANOVA was used for comparison. Differences were considered statistically significant at $P < 0.05$.

## RESULTS

### P4HB protein structure model and sequence analysis

The mRNA and protein sequences of *E. sinensis* P4HB were obtained through RNA sequencing and annotation (Fig. 1). The *P4HB* mRNA sequence was 1,584 bp in length, and the protein had a molecular weight of 58.61 kDa. The protein consisted of 527 amino acids with an isoelectric point of 4.61. To predict the tertiary structure of the P4HB protein, we used the homology modeling method with the human protein disulfide isomerase (PDI) sequence as the template. The predicted tertiary structure model of P4HB is shown in Fig. 2. The similarity between this model and the tertiary structure model of human PDI protein was 58.09%, which exceeded the threshold of 30% and even 50%. These results indicate that the predicted model is highly reliable and meets the basic criteria of homology modeling.

Phylogenetic tree analysis was performed to determine the evolutionary relationships of P4HB in *E. sinensis* with other arthropods. The results showed that *E. sinensis* P4HB protein was clustered with *Portunus trituberculatus*, *Penaeus vannamei*, and *Armadillidium vulgare*, forming one large branch with *Cyphomyrmex costatus*, *Ctenocephalides felis*, *Anopheles stephensi*, *Ctenocephalides felis*, *Anoplophora glabripennis*, *Thrips palmi*, and other arthropods. Meanwhile, *Vargula hilgendorfii*, *Limulus polyphemus*, *Tetranychus truncatus*, and *Araneus ventricosus* were clustered into another group (Fig. 3).

**Figure 1  The *P4HB* mRNA sequence and its protein sequence in *Eriocheir sinensis*.** This figure shows the translation of the P4HB mRNA sequence in *E. sinensis* into the corresponding protein sequence. The length of the *P4HB* mRNA sequence was 1584 bp, and its protein molecular weight was 58.61 kDa. This protein contains 527 amino acids with an isoelectric point of 4.61. Open reading frames in the mRNA sequence are highlighted in red in the figure.     

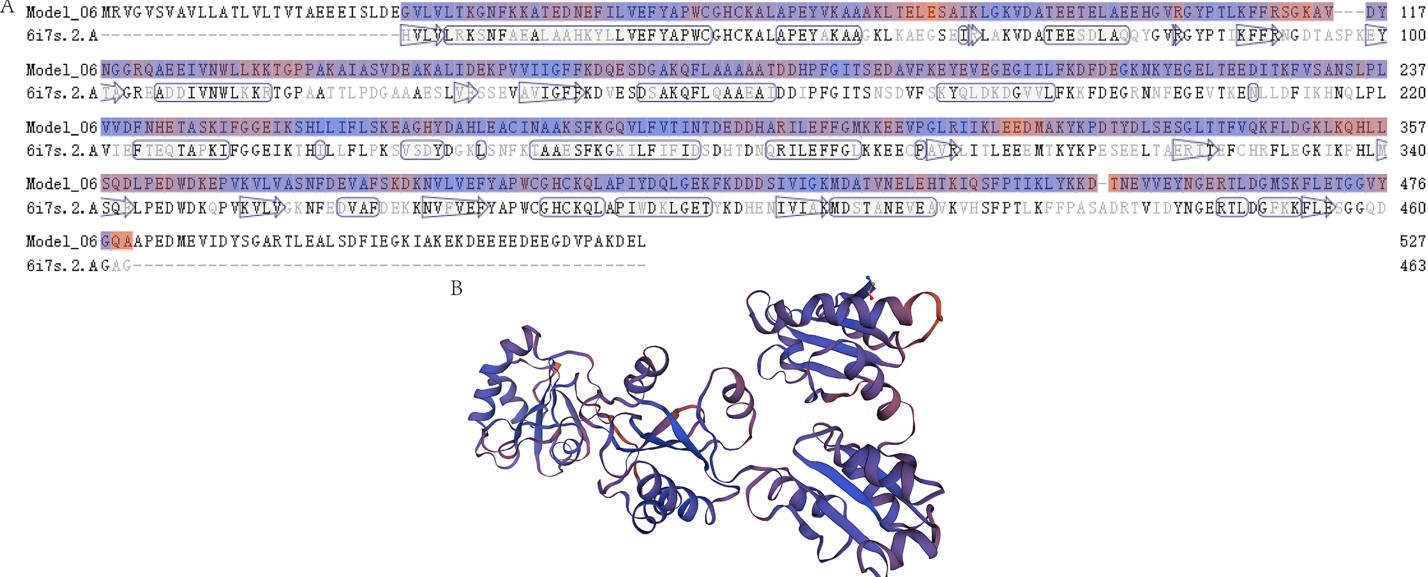

**Figure 2 The prediction of the tertiary structure model of P4HB protein in *Eriocheir sinensis*.** This figure shows the predicted tertiary structure of P4HB protein in *E. sinensis* based on the amino acid sequence of the protein using homology modeling techniques. (A) Sequence alignment between the P4HB protein in *E. sinensis* and the human disulfide isomerase (PDI). The "6i7s. 2. A" represents a protein sequence of human PDI protein with a known spatial structure from the SWISS-MODEL template library (SMTL) which was used as a template for prediction of the tertiary structure model of P4HB protein in *E. sinensis*. "Model_06" represents the amino acid sequence corresponding to the predicted tertiary structure model of the P4HB protein in *E. sinensis*. The arrows in the figure show the direction of alignment between the target and template sequence. (B) A model of the protein tertiary structure of P4HB in *E. sinensis* displayed in the interactive viewer.

It can be observed that P4HB has some degree of species conservation, and the evolutionary relationships of P4HB among different species are relatively conserved.

### Expression of *P4HB* mRNA and protein in testis tissues

The expression of *P4HB* mRNA and protein in the testis tissues of both adult and juvenile *E. sinensis* was detected using RT-qPCR and Western blotting, respectively. β-Actin was used as the internal reference gene and the reference protein. The primer sequences are shown in Table 1. The results demonstrated that P4HB was expressed in both adult and juvenile *E. sinensis* testis tissues, as showed in Fig. 4.

### Expression of P4HB protein in different male germ cells

Immunofluorescence localization was conducted to investigate the expression pattern and subcellular localization of P4HB in male germ cells at different developmental stages in *E. sinensis* testis tissues. The results showed that P4HB was expressed in both adult and juvenile testis tissues, but the expression level in male germ cells at different developmental stages in *E. sinensis* testis tissues was different (Fig. 5). The fluorescence intensity of each cell type, from strong to weak, was spermatogonia, spermatocytes, stage I spermatids, mature sperm, stage III spermatids, and stage II spermatids. There were no significant differences in the fluorescence intensity between the stage I spermatids with mature sperm, stage II spermatids with mature sperm, stage III spermatids with mature sperm, and stage II spermatids with stage III spermatids ($n = 3$, $P > 0.05$). However, there were significant

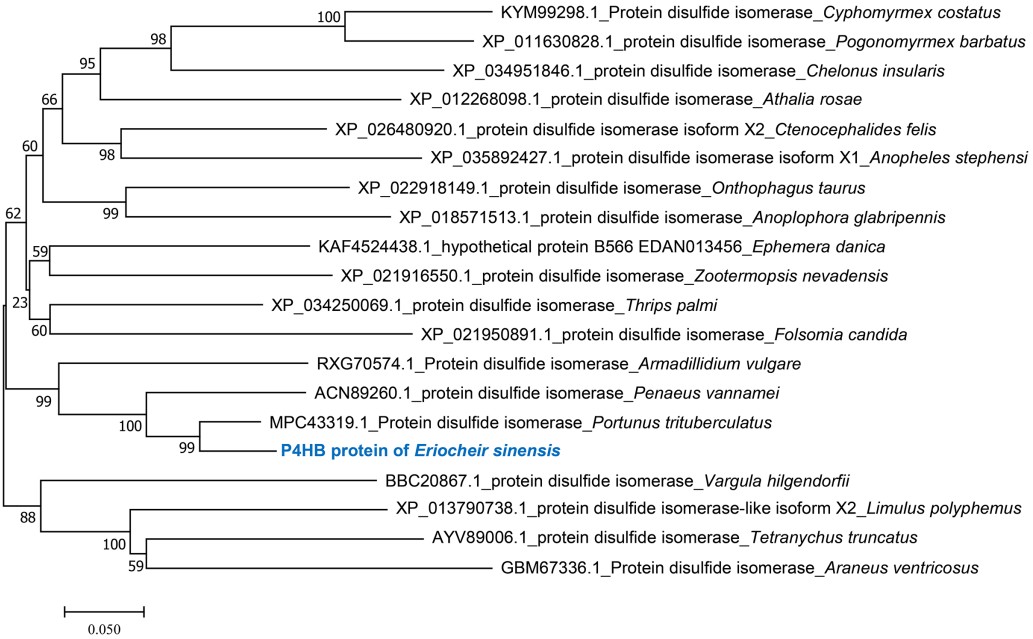

**Figure 3** Phylogenetic tree analysis of amino acid sequence of P4HB protein in *Eriocheir sinensis*. Phylogenic tree was constructed using the neighbor-joining method in the MEGA 7.0 software, based on twenty species' amino acid sequences. Numbers in the phylogram nodes indicate the percent bootstrap, and the bar at the bottom indicates a 5% amino acid divergence in sequence.

**Table 1 The primers of genes used in the study for RT-qPCR.**

| Gene | Forward primer(5′-3′) | Reverse primer (5′-3′) |
|------|----------------------|------------------------|
| *P4HB* | ACAGGCGGTGTCTATGGTCAGG | CCTCGTCCTCCTCTTCCTCATCC |
| *β-Actin* | ACCTCGGTTCTATTTTGTCGG | ATGCTTTCGCAGTAGTTCGTC |

differences between other cell types ($n = 3$, $P < 0.05$). This result showed that P4HB was mainly expressed in spermatogonia, spermatocytes, and stage I spermatids, followed by mature sperm, and was less expressed in stage II and stage III spermatids. Furthermore, P4HB was mainly expressed in the cytoplasm, cell membrane, intercellular substance, and extracellular matrix of spermatogonia, spermatocytes, stage I spermatids, and stage II spermatids. Additionally, P4HB was partially expressed in specific regions of the nuclei of spermatogonia, and it was mainly located in the nuclei and a small amount in the cytoplasm of stage III spermatids and mature sperm.

## DISCUSSION

In most animals, the spermatozoal nuclei are highly compacted and reduced in size to protect the genetic material from external factors before fertilization. However, unlike mammals, the decapod crustacean *E. sinensis* possesses noncondensed chromatin that does not damage DNA (*Wu et al., 2016*; *Sela et al., 2012*; *Hime, Brill & Fuller, 1996*).

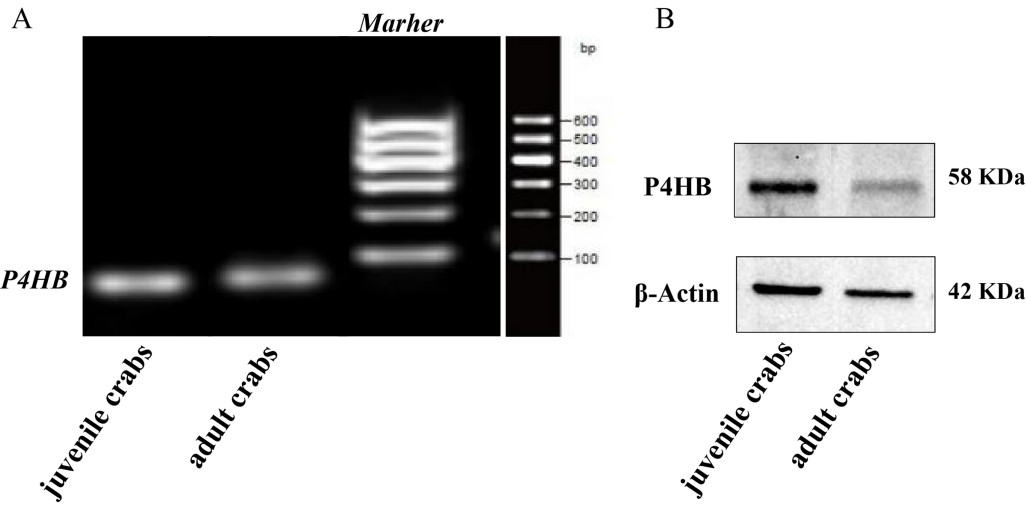

**Figure 4** The expression of *P4HB* mRNA and protein in testis tissues of adult and juvenile *Eriocheir sinensis*. (A) The expression of *P4HB* mRNA in the testis tissues of adult and juvenile *E. sinensis*. The amplified products of RT-qPCR were identified using agarose electrophoresis. (B) The expression of P4HB protein in the testis tissues of adult and juvenile *E. sinensis* using western blotting. The results indicate that P4HB is expressed in the testis tissues of both adult and juvenile *E. sinensis*.

The noncondensed state of the spermatozoal nuclei in *E. sinensis* makes the genetic material loose. Therefore, we conducted a study on the expression and localization of P4HB in testis tissues of adult and juvenile *E. sinensis*. Our study found that P4HB was expressed in both adult and juvenile testis tissues of *E. sinensis*. However, the expression localization varied among male germ cells at different developmental stages.

Previous studies have highlighted the multifaceted roles of P4HB, including catalyzing the formation, breakage, and rearrangement of disulfide bonds, acting as a reductase on the cell membrane surface, and connecting the extracellular matrix to the intracellular skeletal network (*Wan et al., 2020*; *Bi et al., 2011*; *Mezghrani et al., 2000*). Furthermore, P4HB can act as a chaperone protein that inhibits the aggregation of misfolded proteins and facilitates the folding of cytoplasmic proteins (*Yu et al., 2020*; *Bi et al., 2011*; *Lumb & Bulleid, 2002*; *Zhao et al., 2005*). These findings suggest that P4HB plays a crucial role in the synthesis and structural modification of proteins in the cell membrane and cytoplasm. In our study, we observed that P4HB was abundantly expressed in the cell membrane and cytoplasm, particularly on the outer side of the cell membrane and cytoplasm of spermatogonia and spermatocytes. P4HB in these regions can interact with proteins such as integrins, which was closely associated with the actin cytoskeleton (*Kong et al., 2020*). Therefore, P4HB in the cytoplasm can convey extracellular signals and trigger subsequent cascade reactions such as protein phosphorylation and cytoskeletal reorganization (*Ponamarczuk et al., 2018*). Spermatogonia and spermatocytes are in the early stages of spermatogenesis and actively engage in cell proliferation and differentiation. Consequently, coordinated cytoskeletal reorganization is crucial in these stages (*Carlton, Jones & Eggert, 2020*; *Frappaolo, Piergentili & Giansanti, 2022*). Our study discovered

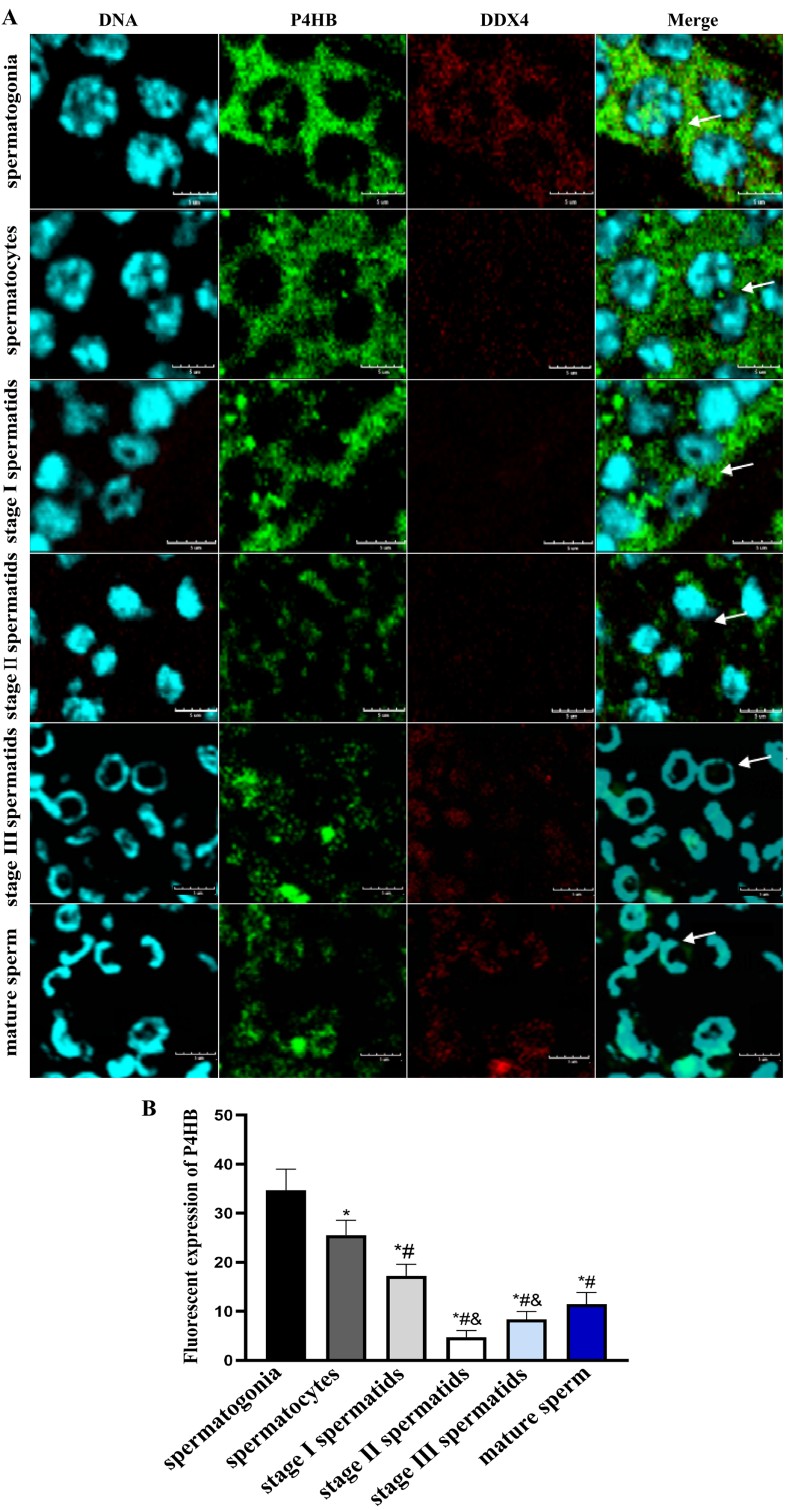

**Figure 5 The expression of P4HB protein in the different male germ cells of *Eriocheir sinensis* (*n* = 3).**
(A) The expression of P4HB protein in the male germ cells at different developmental stages of *E. chinensis*. (B) The comparison of the average fluorescence intensity of P4HB protein in the male germ cells at different developmental stages of *E. chinensis*. The results indicate that P4HB was prominently expressed in spermatogonia, spermatocytes, and stage I spermatids, followed by mature sperm, and was less expressed in the stage II and stage III spermatids. In spermatogonia, spermatocytes, stage I spermatids,

**Figure 5** (continued)
and stage II spermatids, P4HB was mainly expressed in the cytoplasm, cell membrane, intercellular matrix, and extracellular matrix, while in stage III spermatids and mature sperm, P4HB was mainly expressed in the nuclei and to a lesser extent in the cytoplasm, in addition to being partially expressed in specific regions of the nuclei of spermatogonia. DDX4 is a marker protein of male germ cells, primarily distributed in the cytoplasm. The scale of immunofluorescence was 5 μm. * Compared with spermatogonia: *$P < 0.05$; # Compared with spermatocytes: #$P < 0.05$; & Compared with stage I spermatids: &$P < 0.05$.               

elevated P4HB levels in the spermatogonia and spermatocytes, indicating its importance in maintaining cytoskeletal reorganization.

P4HB has been widely demonstrated to play a crucial role in the endoplasmic reticulum, mitochondria, and cytoplasm in various studies (*Zhu et al., 2022*). However, our study showed that P4HB is expressed in specific regions of spermatogonia nuclei, in addition to the cytoplasm, cell membrane, and extracellular matrix. Furthermore, in *E. sinensis* stage III spermatids and mature sperm, it is mainly expressed in the nuclei and to a lesser extent in the cytoplasm. These findings suggest that P4HB's spatial distribution in the male germ cells in *E. sinensis* differs from the common site of P4HB expression in the somatic cells in other species (*Zhu et al., 2022*). Therefore, we speculate that this unusual expression pattern may be related to the mechanism of noncondensed nuclei formation in *E. sinensis*. Unlike mammals, the spermatozoal nuclei in *E. sinensis* undergoes a complex, decondensed process during spermatogenesis (*Chen et al., 2020*), culminating in the unique morphology of its sperm, with flagella-free, microtubule sheaths, and cup-shaped nuclei with radial arms on the surface of sperm (*Sela et al., 2012*; *Wang et al., 2015*). This intricate process is completed in an orderly manner, which requires appropriate protein folding (*Wang et al., 2015*) and the chromatin remodeling (*Wu et al., 2016*) to maintain the structural integrity of the sperm. Therefore, the unique expression pattern of P4HB in *E. sinensis* suggests its potential involvement in ensuring proper protein folding and subsequent sperm development.

A decrease in P4HB can lead to the production of unfolded proteins, reduce cell survival, and result in genomic instability by down-regulating genes involved in DNA repair and cell cycle regulation (*Liu et al., 2019*). The chromatin structure of DNA plays a vital role in genome compaction (*Ou et al., 2017*). Therefore, the high expression of P4HB in the nuclei of *E. sinensis* spermatogonia is likely involved in the formation of its noncondensed chromatin during spermatogenesis. As spermatogonia differentiate, cytoplasmic proteins undergo folding and structural changes, while the nucleus undergoes simultaneous condensation and plasticity changes. Both the later stage of spermatid development and the spermatozoal maturation involve gradual loss of cytoplasm, replacement of histones with alkaline protamine, and repackaging of genetic material in the nuclei (*Wu et al., 2016*). The cysteine sulfhydryl group (-SH) in protamine undergoes oxidation, forming disulfide bonds (-S-S) within and between alkaline protamine molecules in the process, ultimately leading to sperm maturation (*Sharma et al., 2019*). Therefore, the expression of P4HB in the nuclei of the later stages of spermatids and

mature sperm may play a critical role in maintaining homeostasis in the noncondensed spermatozoal nuclei of *E. sinensis*.

In addition, in our study, we observed differential expressed of P4HB in male germ cells at different developmental stages in *E. sinensis*. Previous studies have proved that P4HB can interact with various other proteins in different cellular contexts. For instance, P4HB retained on the surface of Th2 cells can increase disulfide bond reductase activity, altering plasma membrane redox state and enhancing cell migration (*Bi et al., 2011*). P4HB interactions with basic fibroblast growth factor and transforming growth factor in pediatric aortic cells can promote the synthesis of DNA, collagen and matrix proteins (*Fu et al., 2004*). P4HB binding to insulinogen has been found to influence disulfide bonds maturation (*Insook et al., 2019*). Additionally, binding of P4HB to the thyroid hormone T3 can inhibit disulfide bond formation in nascent and misfolded proteins (*Hashimoto et al., 2012*). These findings show that P4HB's diverse functions can be determined by certain factors. Our immunofluorescence localization results showed that the subcellular location and expression of the P4HB protein varied in male germ cells at different developmental stages, which suggest that it is influenced by various factors in its extensive functional activity during spermatogenesis.

P4HB is distributed at high levels in the extracellular matrix or intracellular regions of spermatogonia, spermatocytes, and stage I spermatids and may be involved in spermatogenesis mainly through its oxidoreductase, chaperone, and isoenzyme effects (*Zhu et al., 2022*), which promote the maintenance of the cell's morphology and structure. As spermatids progress from the stage II to the stage III and eventually specialize into the sperm, they discard most of the organelles and excess substances in the cells, further simplifying the cell structure (*Dalton, 2013*). Therefore, the low expression of P4HB in these stages of cells may be due to the reduced requirement of P4HB for its structural maintenance function during this phase of spermatogenesis. Nevertheless, expression of P4HB at appropriate levels during these stages is vital to ensure the proper formation of protamine and the appropriate compaction of sperms, which is critical for successful fertilization. Hence, understanding the role of P4HB during different stages of spermatogenesis is essential for a comprehensive understanding of how this protein regulates spermatogenesis.

Furthermore, our study provided a preliminary analysis of the mRNA and protein sequences of P4HB in *E. sinensis* and found that the protein sequence of P4HB was similar to that of human PDI protein, indicating a similar spatial binding and biological functions (*Kapli, Yang & Telford, 2020*). Phylogenetic tree analysis further revealed that the P4HB sequence in *E. sinensis* shared some similarities and affinities with other crustaceans and arthropods, indicating some degree of species conservation. This conservation of functional domains suggests that different species perform similar functions in the process of evolution. However, different species exhibit diversity in function and phenotype due to the unique environmental pressures and evolutionary adaptations they face (*Sidi, Benjamin & Manyuan, 2013*). This is valuable in the evolutionary analysis of species affinity and can be used as a molecular clock to analyze the homology and evolutionary status of arthropods to a certain extent. This diversity in function and phenotype is also

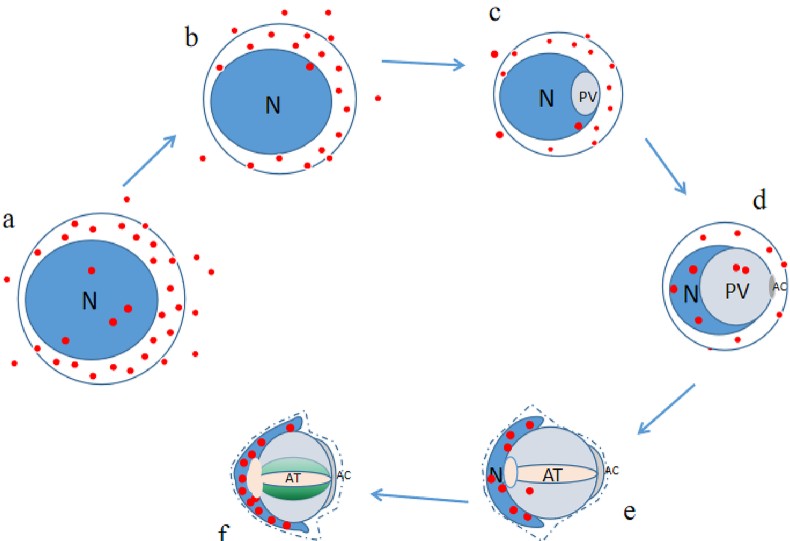

**Figure 6 Expression pattern diagram of P4HB in male germ cells at different developmental stages of *Eriocheir sinensis*.** The red dot in the figure represents P4HB, and each alphabet refers to different developmental stages, *i.e.*, (A) spermatogonia; (B) spermatocytes; (C) stage I spermatids; (D) stage II spermatids; (E) stage III spermatids; (F) sperm, N denotes nuclei, PV represents proacrosomal vesicle, AT represents acrosomal tubule, AC stands for apical cap. The expression and localization of P4HB protein in male germ cells is shown in the figure as follows: (1) spermatogonia: extracellular, cytoplasm, nuclear (some); (2) spermatocytes: extracellular, cytoplasm, nuclei (less); (3) stage I spermatids: extracellular, cytoplasm, nuclei (less); (4) stage II spermatids: cytoplasm, nuclei (more), proacrosomal vesicle (less); (5) stage III spermatids: nuclei (abundant), proacrosomal vesicle (very little); (6) sperm: nuclei (very abundant), acrosome (very little).

reflected in the sequence characteristics of P4HB in different species. Therefore, the analysis of P4HB sequence characteristics and functional domains in different species can help to understand the evolution of species and how it contributes to the diversity of function and phenotype in animals.

Overall, our study deepens our understanding of the role of P4HB in *E. sinensis* spermatogenesis and highlights its potential contribution to the maintenance of cell morphology, stability of spermatozoal nuclei, and genetic stability. This study also provides a foundation for future research on the functional and evolutionary analysis of P4HB in other crustaceans and arthropods.

## CONCLUSION

P4HB was expressed in both the adult and juvenile testis tissues of *E. sinensis*, but the expression level and localization varied among the male germ cells at different developmental stages (as illustrated in Fig. 6). As P4HB plays an important role in the formation, folding, and stabilization of disulfide bonds in proteins, the differences in its expression and localization are likely to be a significant factor in maintaining the cell morphology and structure of different male germ cells of *E. sinensis*. In particular, P4HB expressed in the nuclei of spermatogonia, late spermatids, and mature sperm of *E. sinensis* may play a crucial role in the stabilization of the noncondensed spermatozoal nuclei.

### Funding

This research is supported by the National Natural Science Foundation of China (Grant No. 31960728). The funder had no role in study design, data collection and analysis, decision to publish, or preparation of the manuscript.

### Grant Disclosures

The following grant information was disclosed by the authors:
National Natural Science Foundation of China: 31960728.

### Competing Interests

The authors declare that they have no competing interests.

### Author Contributions

- Yulian Tang conceived and designed the experiments, performed the experiments, analyzed the data, prepared figures and/or tables, authored or reviewed drafts of the article, and approved the final draft.
- Anni Ni performed the experiments, prepared figures and/or tables, authored or reviewed drafts of the article, and approved the final draft.
- Shu Li analyzed the data, prepared figures and/or tables, and approved the final draft.
- Lishuang Sun analyzed the data, prepared figures and/or tables, and approved the final draft.
- Genliang Li conceived and designed the experiments, authored or reviewed drafts of the article, and approved the final draft.

### DNA Deposition

The following information was supplied regarding the deposition of DNA sequences:
The data are available in the Sequence Read Archive (SRA): PRJNA354440.

### Data Availability

The raw data are available in the Supplemental File.

### Supplemental Information

Supplemental information for this article can be found online at http://dx.doi.org/10.7717/peerj.15547#supplemental-information.

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
