# Peer review of "Expression, localization, and function of P4HB in the spermatogenesis of Chinese mitten crab (Eriocheir sinensis)"

_PeerJ, doi:10.7717/peerj.15547_

## Round 0.1 · original submission · Major Revisions

I would have to agree with the reviewers that the current manucript requires major revision in various aspects, below are some of my added comments in addition to those of reviewers:

1. language - kindly proofread the whole document after revision;
2. Method - some parts are still unclear and hinder reproducibility
3. Hypothesis versus statement - As raised by reviewer 3, the current title portrays a bold statement of confirmation, that P4hb IS involved in spermatogenesis and uncondensation of spermatozoal nuclei. However, functional analysis to validate this is insufficient. In addition, discussions are not sufficiently backed by relevant literature.

Reviewer 1 ·

Basic reporting

The English need extensive improvement throughout the manuscript. There are many grammar errors and long and ambiguous sentences.
The research progress on the possible mechanism of uncondensed spermatozoal in decapod crustaceans should be added in the introduction. Why P4hb?

Experimental design

Methods should be descriped in more detail:
1. the antibody dilution ratio in WB experiment for each sample (juvenile and adult testis sample);
2. the calculation of fluorescene intensitiy in different cells is ambiguous,what is the integrated density in the region and the area of the region mean? and the replicates?
3. Appropriate reference should be cited.

Validity of the findings

The results could help better understand the spematogenesis in E.sinesis.

Additional comments

1.Fig.1 is not clear;
2. Apropriate labeling should be added in Fig.3. For example, which is the p4hb sequence of E. sinensis? what is the arrow head/box indicate?
3. The "P4hb" should be in the left of Fig.5A;
4. rotating the labeling of "spematogonia", "spermatocytes"...in the Fig.6;
5. I thought it is over-dicussed concerning the possible function of p4hb on the uncondensed spermatozoal based merely on the protein expression/localization study. Besides, why the expression in stage II spematids is lowest?
6. The discussion of gene/protein sequencing and phylogenetic tree is suggested to put ahead (Line 267-276).

Reviewer 2 ·

Basic reporting

Eriocheir sinensis have uncondensed nuclei in sperm, which is different from the condensed nuclei of most animal sperm. It is important to explore the mechanism of formation and stabilization of the uncondensed spermatozoal nuclei in Eriocheir sinensis. P4hb is a highly abundant multifunctional enzyme, it can catalyze the formation of 4-hydroxyproline in collagen and other collagen-like proteins and plays a key role in the folding of many newly synthesized proteins. This study mainly explored the protein structure and sequence homology of P4hb, and its expression in testis tissue and different male germ cells. Interestingly, it was found that P4hb was mainly expressed in the cytoplasm, cell membrane, intercellular matrix and extracellular matrix in spermatogonia, spermatocytes, stage I spermatids and stage II spermatids, whereas in stage III spermatids and sperm, it was mainly located in the nuclei. This finding may have important implications for spermatogenesis and de-condensation of sperm nuclei in Eriocheir sinensis. This study is innovative and the findings have important theoretical implications.

Experimental design

The experimental design is perfect.

Validity of the findings

The findings in this study is trustable.

Additional comments

1. Figure legends of Figure 5, 6, and 7 are too simple. Describe these figures in detail.
2. Add “Conclusion” part after the “Discussion”.
3. Line 39-40: “The mechanism of formation and stabilization of the uncondensed spermatozoal nuclei in E. sinensis is still not better to be understood.” This sentence needs to be re-written.
4. Line42-43: the setence “After spermatogonia mitosis into spermatocytes, spermatocytes continue to evolve into spermatids, which eventually differentiate into sperm” is not clear. Do you mean after spermatogonia undergo mitosis? Make it clear.
5. Line 50: “The cup-shaped sperm nuclei is an extremely unstable”. Please quote some latest literatures to support it.
6. Lines 51-52: Regarding "Cytoskeleton-associated proteins play a very important role in the specialized sperm nuclei of Eriocheir sinensis", we suggest citing one or two papers to support it.
7. Line 92: Please add a space after the punctuation mark.
8. Line 93: add “Name of City “ before “USA”.
9. Line 123: The relative quantitative value should be the relative quantitative value of P4hb mRNA in the two types of samples, please modify it.
10. Line 143-144: “average fluorescence intensity = integrated density in the region/ area of the region” is suggested to be modified as “Mean fluorescence intensity (Mean) = Integrated Density of the area (IntDen)/Area of the area (Area)”
11. It is recommended to combine Figure 2 and 3 into one figure, and Figure 6 and Figure 7 into one figure, and remove the legend of Figure 7 and mark it at the abscissa.
12. Nomenclature should be checked throughout the entire manuscript, and make sure to name gene/protein correctly.
13. Word spelling should be checked throughout the entire manuscript.

·

Basic reporting

Professional English has been used throughout the manuscript, however there are certain mistakes to be noted of as outline below:
1. Title: The title does not seem to reflect the outcome of the study since there are essentially no functional study that was performed to verify the role of P4hb in spermatogenesis. Most of the functions described were postulated and some postulation lack literature support.
2. Introduction: Line 36: the phrase '......in E. sinensis is still not better to be understood.' is grammatically wrong.
3. Introduction: Line 49-52: The sentence seems twisted and redundant. Please rephrase.
4. Introduction: Line 66-67: What does author mean by ' ........... for healthy reproduction, species reproduction and increased reproduction'?. It sounds ambiguous.

Experimental design

1. Methods: Line 93-94: There is grammatical mistake in this sentence here, please rephrase. Please also provide better details on the RNA extraction, and the sequencing depth and platform used. Not merely mentioning the company name.
2. Methods: Line 96: What does 'according to routine procedures' here mean? The routine procedures for library construction and sequencing data processing etc may varies according to different laboratories. Please provide details into the actual procedures used and also the citation if these methods were modified from previous reports.
3. Methods: Line 97: What does the author mean by generation sequencing? Is it next generation sequencing? If it was, then perhaps the author should provide the details of the platform used and type and depth of the sequencing.
4. Methods: Line 112: Why were the total RNA of only two samples used? Why not a pooled samples to get a better indication of gene expression?
5. Methods: Line 113-114: What does the author mean by 'electric low temperature rapid homogenization'? Please rephrase to clarify or indicate what exactly is the instrument used.
6. Methods: Line 116-119: This long winded sentence here basically describe the western blot. Please just mention where was it adapted from with citation. If it was an in-house unique protocol, please provide the details to each step.
7. Methods: Line 124: How were these primary antibodies derived? Were they derived according to the sequence identification in section 2.2 or was it adapted from previous reports? Please clarify.

Validity of the findings

1. Discussion: Line 190-194: This particular sentence is overwhelming and slightly twisted. Please break it down into several sentences as there are too many points the author is trying to make in this one sentence.
2. Discussion: Line 201-202: The author mentioned that the role of P4hb in spermatogenesis of E. sinensis was actually speculated/ predicted mainly based on the observation of p4hb expression in the outer region of the cell membrane and cytoplasm of spermatogonia, however, they did not provide any further literature to back up their claim. In fact, from Line 202-208, there is a lack of citation overall.
3. Discussion: Line 231-235: What does the author tries to imply with the word 'migration'? Is the author talking about the protein being localized elsewhere in the cell or the changes in protein expression throughout the spermatogenesis in a timeframe? Either way, please provide literature support for this claim. This whole chunk of text here lacks citation supports.
4. Discussion: Line 239-240: Something wrong with this sentence. Sounds like an artifact from a personal comment.
5. Discussion: Line 247- 253: This section here does not seem to relate well with the work describe in this manuscript.
6. Discussion: Line 253-266: This section here is overstretched speculation. The author needs to consider toning down or provide citation for each point. There has to be more valid evidence to support their claim.
7. Discussion: Line 267-276: This whole chunk of text here does not have any citation to its claim which is intriguing as this is the discussion section.

Additional comments

No comments

---

## Round 0.2 · Minor Revisions

I have only major concerns regarding the language (sentences lack coherency, spelling errors, grammatical errors etc) of the whole manuscript. I would suggest the authors to seek professional language proofreading services. In addition, the abstract lacks one or two introductory sentences.

Reviewer 1 ·

Basic reporting

The authors have revised the manuscript carefully and addressed my concern well.

Experimental design

no comment

Validity of the findings

no comment

Additional comments

no comment

Reviewer 2 ·

Basic reporting

The authors have revised the manuscript accordingly, all the concerns raised by the reviewers had been resolved.

Experimental design

Experimental design is good enough.

Validity of the findings

I have no doubt about the validity of the findings.

Additional comments

No further revision is needed for this current vision.

---

## Round 0.3 · accepted · Accept

Please change 'sound limbs' (line 381) to 'complete limbs'. I have no other comments.